# A Modeling and Analysis Study Reveals That CaMKII in Synaptic Plasticity Is a Dominant Affecter in CaM Systems in a T286 Phosphorylation-Dependent Manner

**DOI:** 10.3390/molecules27185974

**Published:** 2022-09-14

**Authors:** Hamish Stevens-Bullmore, Don Kulasiri, Sandhya Samarasinghe

**Affiliations:** 1Centre for Advanced Computational Solutions (C-fACS), Department of Molecular Biosciences, Lincoln University, Lincoln 7647, New Zealand; 2Department of Molecular Biosciences, Lincoln University, Lincoln 7674, New Zealand

**Keywords:** Ca^2+^ signaling, calmodulin, calmodulin-dependent kinase II, protein phosphatase 1, cell signaling, mathematical model, computational modeling, sensitivity analysis

## Abstract

NMDAR-dependent synaptic plasticity in the hippocampus consists of two opposing forces: long-term potentiation (LTP), which strengthens synapses and long-term depression (LTD), which weakens synapses. LTP and LTD are associated with memory formation and loss, respectively. Synaptic plasticity is controlled at a molecular level by Ca^2+^-mediated protein signaling. Here, Ca^2+^ binds the protein, calmodulin (CaM), which modulates synaptic plasticity in both directions. This is because Ca^2+^-bound CaM activates both LTD-and LTP-inducing proteins. Understanding how CaM responds to Ca^2+^ signaling and how this translates into synaptic plasticity is therefore important to understanding synaptic plasticity induction. In this paper, CaM activation by Ca^2+^ and calmodulin binding to downstream proteins was mathematically modeled using differential equations. Simulations were monitored with and without theoretical knockouts and, global sensitivity analyses were performed to determine how Ca^2+^/CaM signaling occurred at various Ca^2+^ signals when CaM levels were limiting. At elevated stimulations, the total CaM pool rapidly bound to its protein binding targets which regulate both LTP and LTD. This was followed by CaM becoming redistributed from low-affinity to high-affinity binding targets. Specifically, CaM was redistributed away from LTD-inducing proteins to bind the high-affinity LTP-inducing protein, calmodulin-dependent kinase II (CaMKII). In this way, CaMKII acted as a dominant affecter and repressed activation of opposing CaM-binding protein targets. The model thereby showed a novel form of CaM signaling by which the two opposing pathways crosstalk indirectly. The model also found that CaMKII can repress cAMP production by repressing CaM-regulated proteins, which catalyze cAMP production. The model also found that at low Ca^2+^ stimulation levels, typical of LTD induction, CaM signaling was unstable and is therefore unlikely to alone be enough to induce synaptic depression. Overall, this paper demonstrates how limiting levels of CaM may be a fundamental aspect of Ca^2+^ regulated signaling which allows crosstalk among proteins without requiring directly interaction.

## 1. Introduction

In the postsynaptic dendrites of neuron, neurotransmission can be adapted to either increase or decrease synaptic conductance. An increase in conductance results in long-term potentiation (LTP) and is associated with learning and memory formation; reciprocally, a decrease in conductance, called long-term depression (LTD), is associated with memory loss [1,2,3,4,5,6,7,8]. Since synapses are malleable in conductance, LTP and LTD are what constitute synaptic plasticity [5,9]. Synaptic plasticity is regulated in both directions by Ca^2+^ signaling.

Ca^2+^ is an important secondary signaling molecule in cells [9,10,11], which binds to four Ca^2+^ ions to activate the protein, calmodulin (CaM) [11,12,13]. Activated CaM can bind and activate over 100 proteins (known as CaM-binding proteins (CaMBPs)) in various cell types [14,15,16]. This allows CaM to regulate many signaling pathways and processes in cells.

In synapses, Ca^2+^-bound CaM can activate both LTP-inducing proteins (usually kinases) and LTD-inducing proteins (usually phosphatases) [11,17,18]. The direction of synaptic plasticity is governed by which set of these CaMBPs receive preferable activation by CaM [9,11,18]. That is, LTD- and LTP-inducing CaMBPs compete to bind available Ca^2+^/CaM [5,9,11]. Understanding the competition between the opposing pathways is therefore important for understanding CaM-dependent memory formation/loss at the molecular level.

Competition among CaMBPs to bind CaM is set by each protein having different binding properties with CaM. Here, competition to bind active CaM is governed by the concentration, on-rate to CaM (i.e., binding velocity) and stability (i.e., dissociation rate from CaM) of each CaMBP with CaM [17,19,20,21]. For example, CaMBPs with a rapid binding velocity may bind to active CaM quicker, whereas CaMBPs with higher stability can bind and become activated with more stability. In addition, the concentration of each CaMBP affects its binding probability with CaM [22]. The nature of Ca^2+^ signaling is what drives competition between LTD and LTP, inducing CaMBPs to bind to Ca^2+^/CaM, thereby affecting synaptic plasticity [11,18].

The molecular number of CaM proteins are outnumbered by the total constituents of its binding targets; this competition for CaM molecules from other molecules has received little attention as a candidate mechanism for tuning CaM signaling.

CaM is a limiting resource that has functional implications for its binding targets [21,22,23,24,25,26,27,28,29]. For example, altering the concentration or affinity of a single protein can affect CaM’s availability to other binding targets [24,25,26,27,28,29]. CaM competition among targets can be understood using mathematical modeling of binding kinetics and concentrations. We can also explore how competition is shaped by perturbing one (local) or several (global) parameters to determine how a competitive CaM binding system behaves. Mathematical/computational modeling can also help make predictions that can otherwise be hindered by human biases/heuristics, which we are vulnerable to do when thinking about complex systems. P. Meehl’s demonstrates that clinicians/psychologists are commonly outperformed by simple mathematical models when making predictions [30]. The use of mathematical modeling in systems biology can thereby serve as a useful tool to make counterintuitive findings. For example, Romano et al. (2017) [21] used computational/mathematical modeling of CaM limiting systems to find that CaM competition could explain the counterintuitive experimental findings of Krucker et al. (2002) [31] and Pak et al. (2000) [32] to understand the competitive binding landscape of CaMBPs with CaM. In the current study, we used mathematical modeling to understand CaM competition between CaMBPs in response to Ca^2+^ signaling in postsynaptic dendrites.

Two of the main proponents of synaptic plasticity are calmodulin-dependent kinase II (CaMKII), which induces LTP, and PP1 which triggers LTD [9,11]. Although PP1 does not bind CaM, its activity is controlled by CaM-dependent proteins. On other hand, CaMKII binds to CaM, which relieves its own autoinhibition. In addition, when two neighboring CaMKII in a 12meric holoenzyme bind to CaM simultaneously, it autophosphorylates residue T286, which increases its affinity for CaM 1000-fold thus “trapping” CaM [33]. In the current study, we used a mathematical model of CaMKII-PP1 system in a CA1 postsynaptic neuron to understand CaM competition when CaM concentration is outnumbered by its binding partners. The model also includes regulation of cAMP. Here cAMP is controlled by CaM-dependent as well as CaM-independent proteins. Within the model, cAMP acts as a key secondary signaling molecule involved in numerous other signaling pathways in all cell types. By modeling cAMP, we studied how the regulation of the two archetypal secondary signaling molecules, cAMP and Ca^2+^, are interrelated.

We studied the transient nature of CaM binding to its CaMBPs to determine how CaM binding occurs during LTD- and LTP-inducing Ca^2+^ signaling events. We also performed a series of global sensitivity analyses, which the process whereby several parameters are varied simultaneously over defined ranges to determine how outputs of the model are affected. This allowed us to understand how the system interacts globally and what are the key underpinning parameters affecting the outputs of the model. The outputs studied in the current paper included Ca^2+^-bound CaM stability and cAMP regulation.

In synapses, Ca^2+^ moves into cells through NMDAR channels as pulses. There are four aspects of NMDAR-dependent Ca^2+^ signaling in synapses: the Ca^2+^ pulse frequency; total Ca^2+^ amplitude; the duration of the Ca^2+^ signal; the location of Ca^2+^ rises [34]. Collectively, these factors determine the direction of synaptic plasticity. For example, in synapses, a high-frequency Ca^2+^ pulse triggers CaM to induce LTP, whereas at a prolonged low-frequency, a low-Ca^2+^-amplitude triggers LTD; low rises in CaM activation lead to preferential activation of LTD-inducing proteins, whereas higher rises lead to LTP-inducing CaMPs receiving preferable activation. However, this cannot be generalized: for example, postsynaptic LTP at cerebellar parallel fiber (PF)–Purkinje cell (PC) synapses induce LTD at high-frequency Ca^2+^ pulses entered via NMDAR channels, which is a reversed mechanism compared to that at hippocampus neurons [35], i.e., low Ca^2+^ frequencies induce LTP in these cell via PP2B. Therefore, a caveat of this model is that it is only limited to the cases where Ca^2+^ high frequencies induce LTP, and low frequencies induce LTD.

Further, Piochon et al. [36] showed the importance of the instructive signals, which are instrumental in the induction of LTP or LTD through the modulation of calcium signals. Based on experiments using Purkinje cells, they showed that the instructive signals originated from climbing fiber (CF) coactivation, strengthening calcium transients and producing LTD independent of the level of frequency stimulation (i.e., 1 and 100 Hz). Calcium buffering gives rise to LTP, but not LTD, at both frequencies, showing that it is the amplitude of calcium that is the critically important factor as the CF instructive signal. However, the absolute calcium amplitudes are not important for inducing LTP or LTD but the LTD threshold slides, which induce LTD at larger calcium transients for a given frequency.

Another limitation of the model is the way we model the influence of PB2B by indirectly associating its influence through Equations (A9) and (A10) to simplify the model, even though PB2B is more relevant in terms of calcium-activated calmodulin. We followed a parsimonious approach towards model building; when this approach is followed to simplify the model, uncertainties naturally become more prominent; hence, our hope was that global sensitivity analyses performed on the model would help us to understand and assess these uncertainties. 

Li et al. analyzed how Ca^2+^ pulse patterns affect CaMKII and PP2B activation [37]. The influence of Ca^2+^ pulse (frequency: 0.5–200 Hz; duration; amplitudes per pulse) were systematically analyzed. According to the results, the binding affinity of I-1 to PP1 was highly sensitive to the frequency of pulses, showing that PP1 activation could make CaMKII a frequency detector. Therefore, the study showed that PP2B can influence the ability of CaMKII to decipher signals despite not directly interacting with it, highlighting the complexity of the system. CaM-mediated I-1 phosphorylation is subject to more complex regulation than PP2B alone. Its counterpart in I-1 phosphorylation, PKA is subject to regulation by CaM-dependent ACs and PDEs. Using a stochastic model, Antunes et al. (2016) investigated Ca^2+^ signaling in the context PP2B versus PKA activation [38]. To model this, the CaM-binding proteins, AC1 and PDE1, which control cAMP were included as well as PP2B. The paper examined competition in the context of kinetic and thermodynamic control (specifically, kinetic control refers to the speed of binding, while thermodynamic control is associated with binding stability). They found that PP2B responded much slower than AC1 and PDE1. PP2B essentially counted the number of pulses, and it showed a pattern of summation of activity as the pulse number increased. AC activated quicker, although it bound to a lesser extent at lower versus higher frequencies; PDE1 activation saturated rapidly at all of the stimulation levels tested. That is, the PDE1 kinetic binding rate gave it preferential activation over other the proteins, especially during short- or low-frequency signaling events. The authors concluded that kinetic and thermodynamic control work together to decipher the Ca^2+^ signal, depending on the nature of the signal (i.e., duration and stimulation frequency). In the model, the PKAc catalytic subunits released to become active at k19f = 0.25 μM^−2^s^−1^ and dissociated at k19b = 0.0016s^−1^, whereas the current investigation focused on Ca^2+^ frequency detection at milliseconds to seconds timescales. If PP2B is able to effectively negate the influence of PKA in a time-independent manner, this shows that PKA is unable to effectively influence PP1 regulation; this, however, may change depending upon the stimulation protocol applied. To include all these interactions would make the model more complex, hence, the parsimonious approach.

Unlike T286 phosphorylation and GluN2B binding, inhibitory phosphorylation plays a reciprocal role and renders CaMKII holoenzymes less responsiveness to Ca^2+^/CaM. As a consequence, CaMKII T305/T306 phosphorylation can alter the thresholds for LTP and LTD [39,40,41,42]. Indeed, there is evidence that increased T305/T306 phosphorylation favors LTD of the synapse while also increasing the threshold stimulus to induce LTP [39,40,41]. For example, Goh and Manahan (2014) found that T305D (a phospho-mimic) mutant mice could not induce LTP at 100 Hz but could in a robust way at 200 Hz; they could more readily induce LTD [41]. It is therefore evident that the phosphorylation state of T305/T306 controls the postsynaptic response to neurotransmission events. Changing the ease by which LTP and LTD are induced is known as synaptic metaplasticity. Another limitation of this model is that it does not include the inhibitory autophosphorylations at T305/306 limiting the model’s ability to explore synaptic metaplasticity.

Interestingly, CaMKII T286D phospho-mimic mutants, becoming less responsive to Ca^2+^/CaM signaling. This lack of response is no longer seen when additional T305A/T306A mutants knocked-in [43,44]. This is because T286 phosphorylation has been shown to facilitate inhibitory phosphorylation [44,45]. This demonstrates how T286 phosphorylation and inhibitory phosphorylation work together to allow for complex regulation of CaMKII and its affinity for CaM. It also shows how aberrant CaMKII phosphorylation can lead to dysfunctional CaMKII dynamics.

This paper is organized as follows: the methods are discussed in Section 2, the results are presented in Section 3, and Section 4 provides a summary and discussion.

## 2. Methods

### 2.1. Ca^2+^-CaM Binding as Ordinary Differential Equations

The N- and C-lobes of CaM have different Ca^2+^ kinetic rates and different forms of binding cooperativity. To model CaM, we used a full 9-state Ca^2+^-CaM model that was developed based on high-resolution (µs) flash photolysis data of the binding kinetics between Ca^2+^ and CaM [46]. This model includes every Ca^2+^-bound state of CaM and will, in turn, help to understand signal detection by CaM.

Kinetic studies phenomenologically describe state transition CaM lobes as transitioning from tense (T) to relaxed (R) states (Figure 1). When Ca^2+^ binds to a lobe, this induces conformational change in the neighboring Ca^2+^ binding site of the lobe, which thereby switches from a tense (T) (closed) to a relaxed (R) (open) state [46]. The cooperativities of the N- and C-lobes work by different mechanisms; the N-lobe has a positive binding cooperativity, i.e., the first Ca^2+^ increases the binding rate of the next Ca^2+^. The dissociation rates in this lobe are relatively high compared with the C-lobe, which facilitates the binding cooperativity by increasing its intrinsic affinity (lower “off-rate”). In general, N-lobe binding is fast but less stable, while the C-lobe is slower but of higher affinity [47].

**Figure 1 molecules-27-05974-f001:**
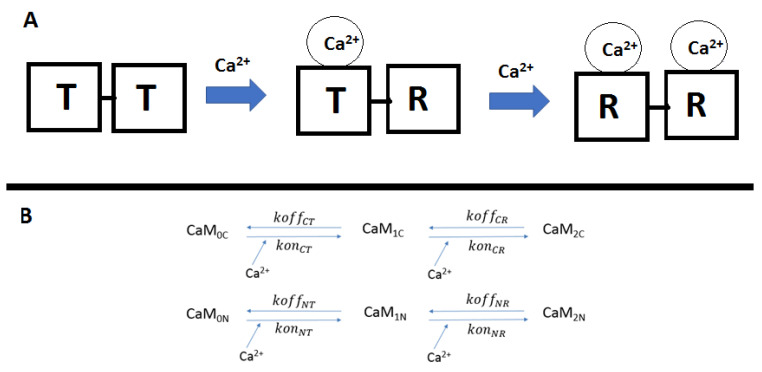
The top figure (**A**) shows how binding of Ca^2+^ hands EF-hands (depicted as black squares) causes the neighboring EF-hand to transition from a tense (T) to a relaxed (R) state. (**B**) Labels above the arrows indicate the parameters pertaining to each reaction. Variables are given in Table 1.

The nine-state CaM model (Figure 2) fed into a larger CaM-binding network of He et al. (2016) [48], which modeled the bidirectional synaptic plasticity of a CaMKII-PP1 system (see Figure 3). In the model, CaM only bound proteins as CaM_4_. The binding partners included AC1 and AC8, which produced cAMP; PDE1 which represses cAMP (and hydrolyzed cAMP to AMP); PP2B; CaMKII (and T286 phosphorylation of CaMKII) (see Appendix A for details of the model and Figure 3). The kinetics of all parameters were taken from He et al. (2016) [48] which were, in turn, taken from experimentally measured kinetics (see Table 2 for the variables used by He et al. in the model and Table 3 for its kinetic parameter values).

The CaM-binding network consists of proteins that regulate PP1 activity and CaMKII. In addition, cAMP is under control by a PKA-feedback loop and CaM-independent AC (denoted as AC*). The activation of PKA is modeled as four states with six parameters as determined by He et al. [48]. Within the context of the current study however, we were most interested in CaM stability. Therefore, we sought to simplify cAMP activation of PKA.

### 2.2. PKA Simplification

In the original model of He et al. [48], PKA activation was modeled as a transition model involving cAMP binding to PKA at the regulatory subunits to release the catalytically active PKAc subunits from the heterodimers. The equations of the model used in this study, based on He et al. [48], are given in Appendix A. Table 2 also gives the initial values of concentration and constants used in the model (see Table 1 and Table 3 in [48] for further details.) The modifications to the original model, based on the results reported in [49], are four cAMP molecules bind to PKA dimers, which results in 2 catalytic subunits becoming released/active in the cell [49]. These reactions involve 6 rate constants and 4 states. In this study we simplified the original equations and modeled PKA activation as a function of available cAMP (See (A26)–(A28) in Appendix A).

PKA [R_2_C_2_] binds to two free [cAMP] molecules at rate k17f, and R_2_C_2_ dissociates from two cAMPs [R_2_C_2_cAMP_2_] at rate k17b. R_2_C_2_ then binds to a further 2 cAMP molecules to form R_2_C_2_ cAMP_4_. Then two catalytic subunits (PKAc) dissociate from the PKA heterodimer at rate k19f.

By simplifying PKA activation as a function of cAMP, we reduce the number of states in the model. In addition, we can also determine whether slow timescales are alone adequate to counteract PP2B, or if PP2B can repress PKA activity regardless of PKA slow activation by cAMP. In the model, the PKAc catalytic subunits were released to become active, as mentioned previously, at k19f = 0.25 μM^−2^s^−1^ and dissociated at k19b = 0.0016 s^−1^, whereas the current investigation focused on Ca^2+^ frequency detection at a millisecond to seconds timescales. If PP2B can negate the influence of PKA in a time-independent manner, this shows that PKA is unable to effectively influence PP1 regulation.

To determine PKA activation as a function of cAMP, the PKA equations from the He et al. model were run until a steady state was reached, and the PKA activation was plotted and from this; the following dose–response Hill equation was fit:(1)$$PKAc=\frac{PKA_{max}}{{\left(1+\frac{IC50}{\left[cAMP\right]}\right)}^{nPKA}}$$

Here, PKAc is the total active PKA concentration; PKA_max_ is the maximum amount of PKA activated; IC50 is the cAMP concentration (µM) at which half of PKA_max_ was active (in µM); [cAMP] is the cAMP concentration (µM); napkin is the Hill constant.

To calculate the parameters of Equation (1), cAMP was run until steady state through a range of values (0–10 µM), which is the maximum range cAMP can reach within the context of the simulations. PKA_max_ was calculated to be 0.2170, and IC50 was 0.3760 µM (the cAMP concentration at which PKAc = 0.1085 µM). Thus, the only variable to be estimated was the Hill constant, napkin. This was calculated using the Curve Fitting Toolbox in MATLAB (https://www.mathworks.com/products/matlab.html). Here, napka was estimated to be 2.569 with an R^2^ = 0.9981.

### 2.3. Ca^2+^ Pulses

Because Ca^2+^ pulses induce synaptic plasticity, we studied the transient behavior of CaM_4_ binding to its targets during the stimulation training and 1 s post-stimulation using the standard model conditions. To apply these Ca^2+^ pulses, Equation (A9) was used in the CaM-CaM binding network. This was performed using the Ca^2+^ pulse patterns 100 Hz, TBS, and LFS, which is described below.
(2)$${Ca}^{2+}={Ca}_{basal}^{2++A\sum_{i=1}^{n}exp^{\left(\frac{-i}{f\tau }\right)}}$$

Equation (2) was developed by Zhabotinsky (2000) [50] and is based on experimental data, which was measured by Ca^2+^ influx into the cytoplasm of a neuron using Fura-2 florescence. This was performed using rat hippocampal slices and was run under physiological conditions at 37 °C and with a pH of 7.2 [51]. Here, $Ca_{basal}^{2+}$ is the basal level of Ca^2+^ in the cell; “A” is the amplitude of Ca^2+^ that fluxes in per pulse; “n” is the number of pulses during the stimulation; f is the frequency of pulses; τ is the decay constant of Ca^2+^ between pulses. Using this equation, Ca^2+^ influx and decay can be modeled without including the mechanisms of Ca^2+^ efflux. If a pulse arrives prior to the previous pulse decays, Ca^2+^ levels summate. Therefore, higher pulse frequencies result in higher Ca^2+^ concentrations (see Figure 4).

The Ca^2+^ inputs applied represented experimental protocols to induce LTP and LTD. The LTP-inducing stimulations included 100 Ca^2+^ pulses per second (100 Hz) as well as theta burst stimulation (TBS) [2,3,50,52]. Figure 4 provides the details. For each of these stimulations, there was a rise in the Ca^2+^ amplitude of 1 µM per pulse. To induce LTD with the Ca^2+^ pulses, we used a low-frequency stimulation (LFS) protocol [4]. This stimulation consisted of a 1 Hz pulse frequency with a Ca^2+^ amplitude of 0.4 µM per pulse.

### 2.4. Mutational Studies

We also sought to determine how T286 phosphorylation shaped CaM competition in a CaM-limiting system. This was chosen because the phosphorylation state increased its affinity for CaM 1000-fold, stabilizing the interaction with CaM_4_, increasing its stability.

PP1 and PDE1 were independently set to zero to determine the local influence on the CaMKII phosphorylation level at each stimulation level. These proteins were chosen because: (1) PP1 dephosphorylates CaMKII directly; (2) of the LTD-related proteins, PDE1 has the highest concentration; therefore, we hypothesized that it may affect CaMKII activation by reducing CaM availability.

### 2.5. Monotonic Plots and Global Sensitivity Analysis

We applied a global sensitivity analysis using Latin hyperbolic sampling (LHS)-partial rank correlation coefficient (PRCC) to understand the global properties of the system for three chosen outputs [53]. First, we analyzed the CaM_4_ concentration, which includes protein-bound CaM_4_, to determine which proteins and catalytic rates stabilized CaM_4_ during Ca^2+^ signaling. Because this output is unaffected by the cAMP simplifications, we used the simplified cAMP model.

Next, using the full model of PKA activation, we analyzed the downstream outputs of CaM signaling, cAMP and I1PP1. These outputs show how Ca^2+^ and cAMP signaling are interrelated, and within the context of a Ca^2+^ signal, how this translates into PP1 regulation. Collectively, by analyzing these outputs, we can understand crosstalk within the system and appreciate the system’s complexity.

For each output studied, we performed an LHS for each input parameter over a range of ± 90% to represent a maximum and minimum range of perturbation [54]. A requirement for PRCC analysis is that each parameter has a monotonic response to perturbation [53]. To ensure this was satisfied, monotonic plots were drawn for each parameter at each of the outputs and for each stimulation. An example is given in Appendix B for TBS stimulation. The parameters that did not display monotonic behavior were truncated to values ranges that were monotonic. Values that had no correlation were removed from the analysis. Table 4 shows the full list of significant PRCC values, and 5000 LHS samples were used per GSA.

We previously saw that CaM4 has low stability during LFS and rapidly dissociates from CaMBPs between each Ca^2+^ pulse. This was reflected by CaMBP concentrations having low sensitivities compared to those at elevated stimulations (see Table 4). In addition, PP1 destabilized CaM4.

Because T286-phosphorylated CaMKII had the highest affinity to stabilize CaMKII, CaM4 stability and the phosphorylation state of CaMKII were positively related. The results show that CaM4 stability was less sensitive to the CaMKII phosphorylation rate (kcat14) (PRCC = 0.17) than by dephosphorylation by PP1 (PRCC = 0.27) and PP2A (PRCC = 0.21). This is because at LFS, CaMKII is sensitive to dephosphorylation by PP1 which is, in turn, regulated by the activities of PKA and PP2B. In the model, PP2B binding to CaM4 was unstable during LFS, whereas PKA was stably activated by cAMP. Consequently, the regulation of PP1 was more sensitive to PKA regulation than PP2B.

Since PKA is controlled by many CaM-independent proteins/reactions, it follows that these proteins were significantly sensitive to CaMKII phosphorylation, and by association, CaM4 stability. In fact, the concentrations of I-1, PP2A, and PDE4D were more sensitive towards stabilizing CaM4 than CaMKII concentration. The most sensitive parameter in the model was PDE1 concentration. Since CaM4 is unstable and not available for long, this means the system is under “kinetic control”, and proteins must compete to bind CaM rapidly, of which PDE1 binds the fastest. It follows that proteins controlling PP1 activity were sensitive. This therefore shows that CaM is largely insensitive to it binding partners but is instead more affected by CaM-independent proteins that control the dephosphorylation of CaMKII.

At TBS and 100 Hz, we saw previously that CaM4 was stabilized by CaMBPs. It follows that the concentrations of CaM targets generally increase. Here, there was a switch from kinetic control to competition of partners being ruled by CaM4’s binding affinity. As a result, the sensitivity of PDE1 reduced; whereas for CaMKII, which binds slower but has a higher concentration, the binding affinity increased in sensitivity and its concentration was the most sensitive parameter in the model.

At HFS, CaMKII acted as a dominant affecter of CaM regulation and bound to the majority of CaM and was even able to downregulate other CaM proteins, including PDE1, in a phosphorylation-dependent manner. In addition, unlike at LFS, CaM4 was more sensitive to the T286 phosphorylation rate than dephosphorylation by PP1 (see Table 4). Therefore, CaMKII is robust to dephosphorylation by PP1. This, coupled with the fact PP1 regulation is dominated by PP2B, means the CaM-independent proteins are not sensitive at HFS.

I1PP1 is controlled by the phosphorylation status of I-1. At LFS, PP2B activation by CaM was unstable. Consequently, the PP2B concentration had no significant effect over I1PP1. Meanwhile, PKA activity was stably activated by cAMP. As a result, there was much overlap among the parameters affecting cAMP and I1PP1.

Due to the instability of CaM, its binding targets AC1/8 and PDE1 had low sensitivity towards cAMP levels. Instead, the influence of these parameters was masked by CaM-independent ACs, (collectively called AC*). Here, the concentration of AC* was the most sensitive parameter to both cAMP and I1PP1. Because LFS is a prolonged stimulation protocol, and PKA is slow to activate, this affords PKA time to activate and further activates PDE4D. PDE4D then represses cAMP levels. This was reflected in the PRCC results (see Table 5), whereby each of the binding steps of cAMP to PKA were significant, as was PDE4D concentration. In fact, PDE4D was the second most sensitive protein to both cAMP and I1PP1. Overall, during LFS, cAMP and I1PP1 were sensitive to both CaM-BPs and CaM-independent proteins, and the number of sensitive parameters was higher at LFS versus HFS (see Table 5 and Table 6).

At the HFS of TBS and 100 Hz, PP2B stably binds to CaM4. Because PP2B activation by CaM4 is quicker and has higher total activity than PKA; this means PKA phosphorylation of I-1 becomes offset by the dephosphorylation of PP2B. It follows that PP2B negates the effect of PKA and the cAMP modulators with respect to I1PP1 formation. This also means that the crossover between cAMP regulation and I1PP1 is lost, and there is disparity between the GSAs of I1PP1 and cAMP. That is, the number of sensitive parameters towards I1PP1 formation is reduced, since cAMP and PKA parameters are no longer sensitive.

Despite cAMP having little influence over I1PP1 levels in the cell, it is still an important secondary signaling molecule across all cell types and is involved with numerous processes, which act upon different timescales. Although studying this output within the context of HFS has little influence over PP1 regulation, studying it can nonetheless help to understand crosstalk between Ca^2+^ and cAMP signaling, which are the two main secondary signaling molecules in cells.

At TBS and 100 Hz, CaM4 is stabilized and, as a result, cAMP regulation switches from being primarily regulated by CaM-independent proteins, to being controlled by CaM-binding proteins that regulate cAMP. The concentration and catalytic activity of AC1 are the most sensitive parameters towards cAMP, which is three-fold more sensitive towards cAMP levels than its opposition, PDE1, despite binding to less CaM4.

We have seen that CaMKII represses AC1/8 and PDE1 during HFS. Since AC1 is more sensitive to cAMP than PDE1, this means that CaMKII repression of these targets has a net negative effect on cAMP levels. At TBS, CaMKII is more sensitive than PDE1 over cAMP; at 100 Hz, CaMKII becomes the most sensitive CaM-binding partner due to the fact of its downregulation of AC1 (see Table 4). In fact, at 100 Hz, CaMKII concentration is the most sensitive parameter to reduce cAMP levels despite not acting directly upon cAMP. The influence of CaMKII over cAMP is also reflected in the I1PP1 results. In fact, at 100 Hz, CaMKII negates the influence of cAMP proteins such that it is the only protein which regulates cAMP to affect I1PP1 formation.

Overall, because CaM-independent proteins are less sensitive when CaM is stabilized, the total number of sensitive parameters at HFS are reduced. This is especially evident regarding I1PP1, whereby the influence of cAMP-regulating proteins get negated by PP2B. Indeed, at LFS there were 35 sensitive parameters, whereas at 100 Hz, only seven parameters were associated with CaMKII and PP2B (see Table 5).

The statistical analysis employed is extensively discussed in [48] within the context of model building based on a more thorough review given by Mariano et al. [53]. For completeness, we provide a very brief summary of the general procedure here. We subjected the kinetic parameters of the model to GSA by regionalized sensitivity analysis with the predefined range of variation for each parameter: ±90% of the baseline value. Then, we generated 50,000 sets of parameters so that each set was a parameter vector randomly chosen from the corresponding predefined range using LHS, because LHS gives simultaneously varied, evenly distributed, and minimally correlated samples. We ran the model using the 50,000 sets of parameters, one set at a time. A set of parameters is acceptable if the set produces the expected behavior of the model [48,54]. Then, we investigate the sensitivity of a parameter by comparing its “acceptable” and “unacceptable” cumulative frequency distributions (CFDs) from the corresponding parameter sets using Kolmogorov–Smirnov (KS) test. A sensitive parameter must have a sensitive range (a subset of the predefined range) such that the samples within this range should have a different probability of being accepted, compared to other ranges. The probability of being accepted for a particular parameter within a subrange can be approximated by the acceptable proportion of all the parameter sets which have the particular parameter within the subrange. On the other hand, an insensitive parameter has a consistent probability of being accepted over the predefined range; the probability of being accepted for any subrange within the predefined range equals the probability of being accepted for the predefined range. The statistical difference between the two CFDs was evaluated by the KS test [48]; the *p*-value calculated by KS test determines how likely (between 0 and 1) the two distributions are the same, and we defined a *p*-value less than 0.01 to be the “significantly different” interval. PRCC is a robust sensitivity measure for nonlinear but monotonic relationships between an input range to an output as long as little to no correlation existed between the inputs. The identified ranges were used to calculate the PRCC values, and corresponding *p*-values were calculated using the procedures given by Marino et al. [53].

## 3. Results

In the current study, we combine a 9-state ODE model of CaM activation with a larger model of CaM-regulated bidirectional synaptic plasticity [46,48]. Using this model, we determined the influence that Ca^2+^ signaling had between CaM and its binding partners. Specifically, we investigated how limiting CaM levels shaped competition of CaMBPs for CaM_4_. Here the competition among CaMBPs was governed by the kinetic binding properties (i.e., binding and unbinding rates with CaM_4_) and concentration of each target. The kinetics of all parameters were taken from [48] which were, in turn, taken from experimentally measured kinetics (see Table 3). The model thereby makes predictions as to systems behavior, which are founded upon results by experimental research groups. We first discuss the transient behavior of the model when Ca^2+^ pulses are applied under standard conditions.

At HFS, CaM rapidly bound to CaMBP, which increased the stability of CaM_4_ [19]. Figure 5 shows that the total pool of CaM converted into CaM_4_ during TBS and 100 Hz. This stability of CaM_4_ was facilitated when it bound to CaMBPs [19]. Consequently, as CaM availability was saturated by protein binding, CaMBPs had to compete for this limiting resource.

After CaM becomes saturated by binding CaMBPs, each protein target competed for the limiting CaM pool. As Figure 6 and Figure 7 show, the nature of this competition changed when the stimulation progressed. Initially, AC1/8 and PDE1 bound to CaM, and the concentration of the CaM-bound forms of these proteins increased. As the stimulation progressed, the concentration of these proteins reduced, whereas the CaMKII and PP2B concentrations bound to CaM continued to rise. This occurred because CaM is redistributed away from AC1/8 and PDE1, which although have fast binding kinetics have relatively low affinity and dissociate from CaM_4_ between Ca^2+^ pulses. CaMKII, however, remains stable, and its CaM-bound states continue to increase. This means that available CaM is redistributed from AC1/8 and PDE1 to bind CaMKII instead. Since AC1/8 and PDE1 require CaM_4_ binding for activation, this means CaMKII effectively represses the activation of these proteins. In this way, when there is an elevated Ca^2+^ stimulation, CaMKII may act as a dominant affecter over CaM signaling in a competitive CaM-binding network.

During LFS, CaM_4_ is unstable and rapidly decays between each Ca^2+^ pulse. Of the CaM targets, PDE1 binds to the most CaM, as it has the fastest binding kinetics. Evidently the competition of targets during LFS differs that from at HFS and is governed by “kinetic control”. Here, CaMBP concentration and binding speed are the most important factors that shape CaM binding competition. It is noteworthy, however, that since CaM_4_ is unstable, none of the CaM binding targets have stable interactions with CaM_4_ (see Figure 8). This means that under the assumptions of the model, CaMBP activation by CaM_4_ is unstable during LFS.

At LFS, phosphorylated CaMKII is the least stable CaM-bound state in the system. This is due to the fact of dephosphorylation by PP1 and PP2A. PP1 activity is, in turn, controlled by the balance of PKA and PP2B. The results show that CaM_4_ activation of PP2B during LFS is unstable, whereas PKA is stably activated by cAMP. This means that during LFS, PKA is the main regulator of PP1 activity which dephosphorylates CaMKII. In addition, PKA has its own feedback loop, whereby it activates PDE4B/D which repress its activation. Because PKA and PDE4B/D regulate PP1 and, thus, the phosphorylation status of CaMKII, PKA and PDE4D thereby control the amount of CaM_4_ that is stabilized during LFS (Figure 9).

Because T286 phosphorylation of CaMKII increases its affinity for CaM 1000-fold, we hypothesized that the redistribution of the CaM pool was T286 phosphorylation dependent, as it would lend CaMKII a competitive binding advantage for CaM binding. To study this, we set the T286 phosphorylation rate to zero, analogous to a T286A knock-in mutant [55].

Figure 10 shows that in the absence of T286 phosphorylation, CaMKII no longer represses low-affinity CaM targets. This demonstrates that CaMKII acts as a dominant affecter to downregulate CaM binding targets in a phosphorylation-dependent manner. The model thereby provides a novel insight as to the functionality of T286 phosphorylation during HFS. In addition to T286 phosphorylation, CaMKII reduces the availability of CaM due to the fact of its relatively high concentration compared to other binding partners.

Next, we tested the effect of PP1 over CaMKII phosphorylation at the HFS and LFS protocols. Here, Figure 11 show that at HFS, CaMKII is robust to PP1 activity, whereby knocking out PP1 showed a minor influence over its phosphorylation status. Reciprocally, knocking PP1 out during LFS led to an accumulation of phosphorylated CaMKII throughout the entirety of the stimulation (see Figure 11).

Given the robust nature of CaMKII to PP1 at HFS, we sought to determine how PDE1 concentration activity may affect CaMKII phosphorylation. This was performed because PDE1 may compete with CaMKII at elevated Ca^2+^, since other than CaMKII, it bound to the most CaM_4_.

Figure 11 shows that that knocking out PDE1 was more influential over CaMKII phosphorylation than knocking out PP1, despite PP1 acting directly upon CaMKII. This was because PDE1 competes with CaMKII for CaM and, in this way, could downregulate CaMKII without the requirement to interact with it directly.

## 4. Summary and Discussion

In the current study, we tested whether limiting CaM levels affected Ca^2+^ signaling in synapses. To do this, we used mathematical modeling of a CaM-binding network associated with synaptic plasticity induction. Since the CaM concentration was lower than the combined constituents of its binding partners, this meant CaM-binding partners competed for available CaM. Using this model, we investigated competition among CaMBPs for CaM during both high- and low-frequency Ca^2+^ signaling. We also investigated the crosstalk between cAMP and Ca^2+^, as the model contained CaM-dependent cAMP affecter proteins, AC1/8 and PDE1. The model predicted that limiting CaM levels can shape CaM-dependent signaling networks associated with synaptic plasticity and may be an important aspect of signaling networks that can give rise to feedback loops and additional levels of control in networks.

Our results support the notion that limiting CaM levels can affect Ca^2+^ signaling and cause cross-regulation between binding partners during HFS. It creates competition, and through this competition arises feedback loops and crosstalk among proteins in CaM-binding networks. We showed that at HFS, the total available CaM rapidly binds to its binding partners, which is in-line with experimental evidence [26]. We found that during HFS, there was an initial rise in the CaM-bound portions of AC1/8 and PDE1. As the signal progresses, the total pool of CaM became saturated to binding CaMBP targets. CaMBPs must thereafter compete for the limiting number of CaM molecules that are available. It follows that CaM was redistributed among its binding targets. Here, CaM redistributed from AC1/8 and PDE1 to bind to CaMKII instead. This redistribution was mediated by the T286 phosphorylation of CaMKII, which reduced its dissociation rate with CaM by 1000-fold [33]. Since AC1/8 and PDE1 have lower CaM-binding affinities, they dissociated from CaM between Ca^2+^ pulses, whereas CaM retained stable binding to phosphorylated CaMKII. Since these CaMBPs require CaM binding to gain full activation, this means CaMKII represses AC1/8 and PDE1 and acts as a dominant affecter of CaM-activated networks. In this way, CaMKII can effectively repress other unspecific low-affinity CaMBPs by disallowing those proteins access to CaM which would otherwise activate them.

Because CaMKII represses cAMP affecters, this also affords CaMKII to indirectly regulate cAMP levels. The GSA results revealed that cAMP was three-fold more sensitive to AC1 than PDE1. It follows that CaMKII repression of AC1/8 and PDE1 led to a net decrease in cAMP levels. This also demonstrates how CaMKII can act as a dominant affecter in CaM-regulating networks and allows for additional mechanism by which crosstalk between cAMP and Ca^2+^ occurs. Overall, T286 phosphorylation, therefore, gives CaMKII additional functionality to downregulate unspecific low-affinity CaM partners during Ca^2+^ signaling.

T286 phosphorylation has been postulated to allow CaMKII to remain autonomous and remain bistable after the signal has passed [56,57,58]. Experimental evidence has shown, however, that CaMKII rapidly becomes deactivated within 1 min after the signal has passed [59]. In the current work, we showed that increased binding affinity plays a role during elevated Ca^2+^ signaling, an aspect never before postulated in the literature. Although this has not been tested, in a similar study, Rakhilin et al. (2004) [27] experimentally showed that the protein “regulator of calmodulin signaling” (RCS) can also downregulate other CaMBPs in the system in a phosphorylation-dependent manner. They showed that phosphorylated but not unphosphorylated RCS could downregulate other CaMBPs without directly interacting. Overall, the current study thereby postulates that T286 phosphorylation of CaMKII has intratrain functionality to repress other CaMBPs of CaM-binding networks.

In the model, we also found that because CaMKII can modulate cAMP affecters which, in turn, regulate PKA, we showed a counterintuitive regulation between the two LTP-inducing kinase proteins, PKA and CaMKII. This coregulation arises because CaMKII indirectly represses cAMP levels, meaning it also represses PKA activation by cAMP. We propose, here, that this may act as a feedback loop. That is, since both PKA and CaMKII can induce LTP, if CaMKII binding to CaM is restricted, there would be an increase in cAMP and PKA activation.

It has been shown in cardiac cells that inhibiting CaMKII from binding CaM leads to an increase in cAMP production (and thereby, increased PKA activation) [60]. It is worth noting, however, that the study showed that this was due to the catalytic activity of CaMKII and not due to CaM availability. Ca^2+^ is regulated differently in cardiac cells and different pathways of CaM regulation occur. The study does, however, link cAMP and Ca^2+^ regulation and makes the inference that CaMKII and cAMP have evolved to be coregulated. The authors of the study supported the notion that crosstalk between the two kinases, CaMKII and PKA, may be an important form of feedback in the system.

The current model shows that limiting CaM levels can shape CaMBP regulation. The occurrence of limiting CaM numbers having functional consequences have been previously postulated [22], and recently modeling work has suggested that it may give rise to counterintuitive forms of regulation, whereby unrelated molecules may affect the activation of other members of the CaM binding pool. Romano et al. (2017) [21] showed that limiting levels of CaM can explain unexpected experimental results and have also suggested that limiting CaM can give rise to feedback loops. Limiting CaM levels allows CaMKII to regulate CaMBP activities without requiring interacting with those proteins directly. Given that CaMKII functions to transduce Ca^2+^ signaling in multiple pathways across many cells and given that autophosphorylation of CaMKII is universal in CaMKII regulation, this affords CaMKII to act as a dominant affecter regardless of cell type when CaM dependent signaling occurs in response to elevated cellular Ca^2+^. Indeed, CaMKII is an abundant protein in the brain, constituting 1–3% of total protein in the forebrain of rodents and acts in many regulation pathways and processes [61,62,63,64,65].

Although the current model is focused on the dendritic spine, the role of T286 phosphorylation downregulating other proteins may be an evolutionary mechanism by which CaMKII can regulate other CaM-binding proteins without the requirement to interact directly with them, per se. For example, CaMKII has been shown to regulate a variety of cell types such as those in skeletal muscle ([66] sciatic nerves and pain pathways [67,68]), cardiac cells [69] (and has been shown to play a role cardiac arrhythmia, cardiac hypertrophy, and heart failure [70,71]); bone [72]; pituitary cells [73]; PC12 cells [74]; cerebellar cells [75]; cell differentiation (and is consequently involved in cancer of many other cell types [76]). Given the wide variety of processes and cell types CaMKII operates under, it is feasible that autophosphorylation acts as a universal mechanism to allow CaMKII to act as a dominant affecter and downregulate other unspecific CaM-activated proteins in these systems.

In addition, the drugs KN-62, 92 and 93 prevent CaMKII binding to CaM. The current study suggests that use of this class of drugs would lead to upregulation of all other low affinity CaMBPs in the cell, regardless of cell type. For example, Brooks and Tavalin (2010) [77] found that the use of the inhibitor KN-93, leads to changes in affinity for other CaM binding proteins; Mika and Conti (2015) found that applying KN-93 lead to upregulation of cAMP. In addition, KN-92 has been applied to reduce cell proliferation in cancer rodents [76]. Future experiments could test the prediction, although there may be other factors at play which could alter these predictions. For example, the model does not include spatial aspects of CaM signaling such as diffusion rates. Moreover, the current model is deterministic, and stochastic elements may affect CaM signaling networks. Neurotransmission, for example, is a probabilistic process, and the neurotransmitter released by the presynaptic neuron has probabilistic aspects that are important for tuning responsiveness of the postsynaptic release of Ca^2+^.

An aspect of CaMKII regulation not covered in the current model is inhibitory phosphorylation. We can, however, make predictions as to the effects it could have. The role of inhibitory phosphorylation is to prevent CaMKII from binding to CaM_4_. The current study suggests that this would lead to upregulation of other CaMBPs. Indeed, mutant studies which have inhibitory phosphorylation phospho-mimics have shown a shift in “metaplasticity”, i.e., the synapses with these mutants require higher levels of stimulation in order to activate LTP [40,42]. What is important to consider, however, is that LTP is still able to be induced, just as it can be induced in T286A mutants [78]. A reason LTP is still possible is that AC1/8 may be able to produce more cAMP in the cell. This would thereby increase the activation of PKA which is also able to induce LTP.

One example of pathology arising from excessive inhibitory phosphorylation has been noted in a disease called Angelman syndrome, a form of mental retardation [43]. Perhaps the pathology of this disease arises due to dysregulation of other processes such as cAMP regulation and other pathways getting triggered improperly, although it is possible that this may in fact have mechanisms to compensate for lack of CaMKII functionality, aberrant phosphorylation would restrict the synapse to exert the properly fine-tuned response in which the synapse has evolved to exhibit. Therefore Ca^2+^ signaling in the synapse would be disrupted leading to the cell having faulty LTP and/or even LTD.

Another potential implication of the current study is that CaM have been shown to decline with age [79]. Our results predict that a reduction in CaM levels may alter CaM-competition and lead to dysregulation of CaM binding networks. For example, low-affinity CaMBPs may get buried by other proteins with higher concentrations and/or higher affinities. GluN2B binding to CaMCaMKII^p^ enhances trapped CaM affinity for CaMKII 20-fold further which could help CaMKII to also coordinate spaced apart trains as well [80,81,82]. Because this high affinity, CaMKII may be able to remain bound longer and thereby more effectively prime CaMKII for activation in response to future trains. Also, since its affinity for CaM increases, GluN2B-CaMKII could repress binding of other targets by shielding CaM from binding to other targets. In addition, GluN2B binding protects T286 phosphorylation from PP1 [83,84]. Indeed, adding GluN2B binding may play an important role in CaM competition and this role could be studied experimentally and/or with modeling studies.

Another consequence of CaMKII’s ability to regulate other CaM-targets is, that could be when the commonly used CaMKII inhibitors KN-62, KN-92, and KN-93 are applied to systems. The mechanism of action for these inhibitors is to block the CaM-binding region of CaMKII [85]. Studies have used this class of inhibitors to ascertain CaMKII function [85]. The current model, however, suggests that some results from these studies may arise due to upregulation of other partners: other proteins may bind to more CaM, since more would be available.

A progression of aging is a reduction in CaM production which, in turn, is associated with a decline in synaptic function [79,86]. This means understanding how CaM availability affects cell signaling may shed light as to how CaM may affect other pathways and/or regions of the brain. Lowering CaM may affect LTD related proteins due to CaMKII repressing their activities; alternately, less CaM leads to a bias of the CaM bound fraction towards the fast binding LTD related proteins if redistribution does not occur. An additional possibility is that reducing CaM concentration could reduce CaMKII phosphorylation. This is because excess CaMKII reduces the probability of T286 phosphorylation due to a lower probability that both the acting substrate and catalytic subunits of the holoenzyme are CaM bound [87].

Modeling the effect of limited CaM availability can also be applied to a current avenue of cancer research. There is a potential cancer treatment being trailed which uses the drug ophiobolin A [88]. This drug reduces CaM availability as CaM is implicated in over activation of the oncogene KRAS which can promote cell proliferation [88,89,90]. Although this protein network is not part of the synaptic plasticity pathway, the CA3 → CA1 synapse can serve as a model to predict how reduction in CaM may lead to improper regulation of other proteins in the CaM binding system. Our model predicts that low-affinity CaM targets would get masked when CaM levels are reduced.

## Figures and Tables

**Figure 2 molecules-27-05974-f002:**
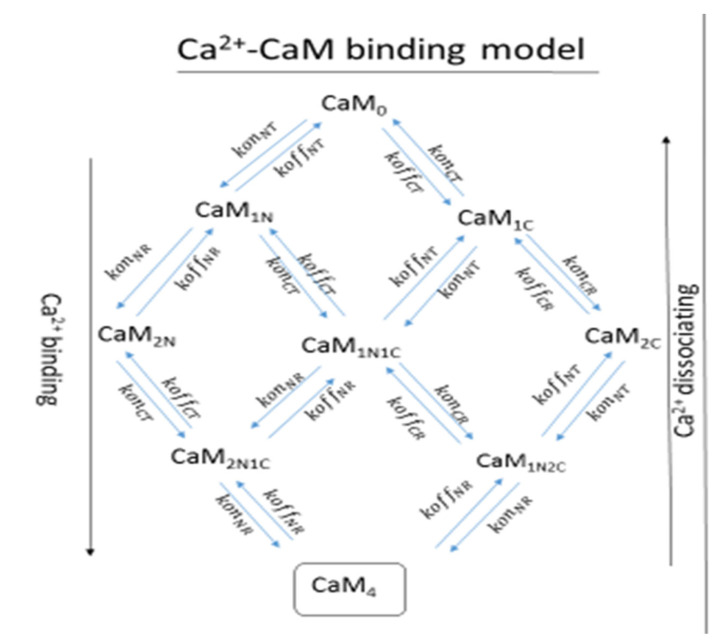
A schematic diagram of the reactions of the model used. On the left is the Ca^2+^-CaM binding model. Here, CaM_o_ represents CaM bound to no Ca^2+^; CaM_1N/2N/1C/2C_ represent 1 and 2 Ca^2+^ ions bound to the N- or C-lobe, while CaM_4_, the output of the model represent, 1 and 2 Ca^2+^ ions bound to the N- or C-lobe, while CaM_4_, the output of the model, represents Ca^2+^ binding and dissociation, respectively, with the subscripts after representing either the T or R state in which the Ca^2+^ is binding or unbinding.

**Figure 3 molecules-27-05974-f003:**
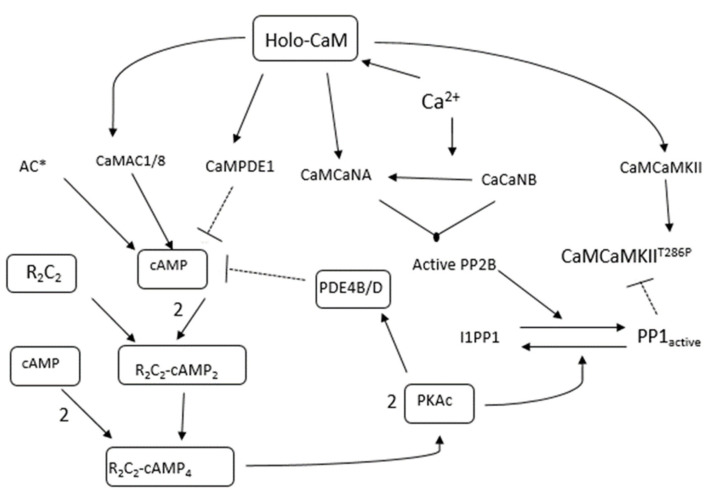
Pathway diagram representing reactions used in the mathematical model used in this paper. All arrows represent mass action reactions (all of which are reversible). Here, upon the binding activation by Ca^2+^, Holo-CaM can bind to and activate AC1/8, PDE1, PP2B subunits (i.e., CaNB and CaNA), and CaMKII. Upon binding to CaMKII, CaMKII can phosphorylate subunits of neighboring CaMKII subunits within a 12meric holoenzyme. The schematic diagram also includes complex regulation of PP1. PP1 is regulated by PKA and PP2B. This is because PKA and PP2B phosphorylate and dephosphorylate I-1, respectively. When I-1 is phosphorylated, it binds to and inhibits PP1. When I-1 is dephosphorylated by PP2B, it dissociates from PP1, and PP1 is active. PKA is activated by cAMP which, is in turn, controlled by ACs and PDEs. Upon activation by cAMP, as well as phosphorylating I-1, it can also phosphorylate PDE4B/D, which then represses cAMP, CaMKII, PP1, and PKAc; activates PP2B; then coordinates the phosphorylation sites of AMPAR. Phosphorylation of AMPAR can modulate its conductance and numbers anchored in the PSD.

**Figure 4 molecules-27-05974-f004:**
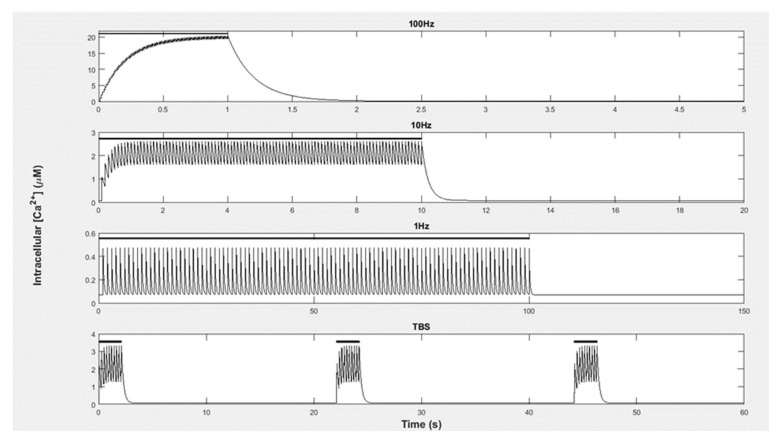
The Ca^2+^ level in response to the stimulation trainings (as indicated by the black line above the *y*-axis) to 100 pulses per second (100 Hz), 10 pulses per second (10 Hz), and one pulse per second for LFS (1 Hz) and the more complex training of the TBS pulses: Each of the 3 trainings consisted of pulse pattern consisting of 10 epochs. Each epoch contained 4 pulses separated by 10 ms (100 Hz) and each epoch of the training was separated by 0.2 s. Furthermore, each training was separated by 20 s. In each of the stimulations, it can be observed how the Ca^2+^ decayed between each pulse and, as a consequence, if the pulses were closer together, then the Ca^2+^ could accumulate before the previous pulse decayed. At LFS, each pulse almost completely decayed between each stimulation and, therefore, the Ca^2+^ did not accumulate.

**Figure 5 molecules-27-05974-f005:**
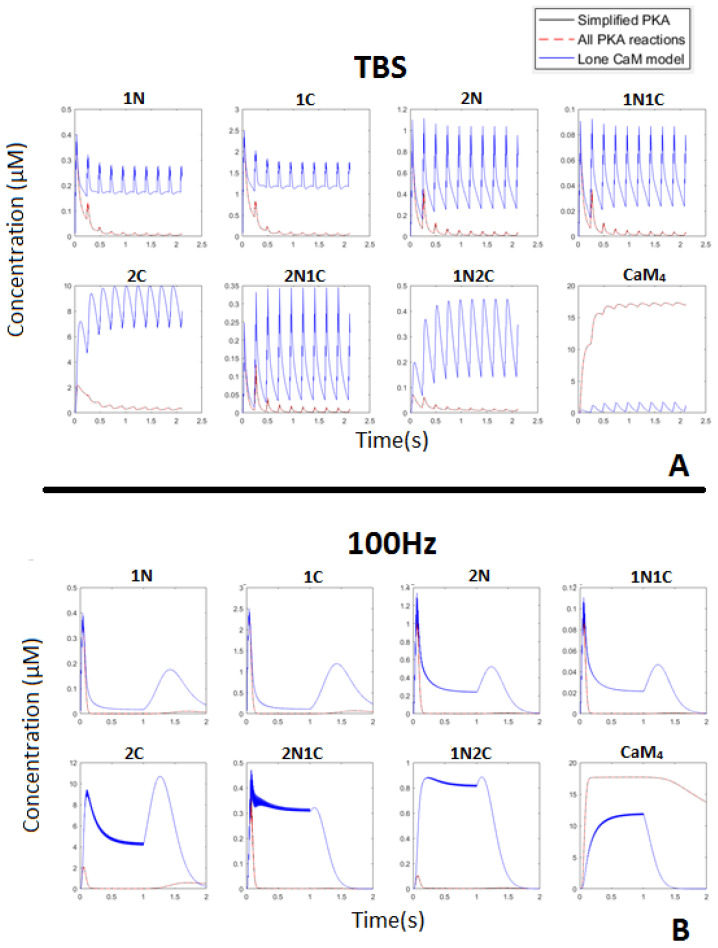
CaM states during the TBS (**A**) and 100 Hz (**B**) stimulation. Notice that each partially bound state’s peak rapidly decayed in the full CaM-binding system, unlike in the lone CaM model, where calcium bound to either single or multiple sites (i.e., <4). This was because the stability of bound CaM increases when all the sites are bound to Ca^2+^, i.e., partially bound CaM trends towards instability.

**Figure 6 molecules-27-05974-f006:**
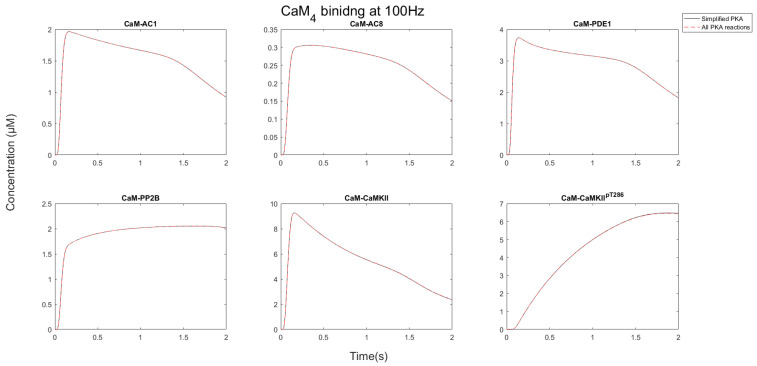
Quantity of CaM-bound targets during 100 Hz stimulation. Notice that AC1/8 and PDE1 peak and then decrease as CaM_4_ is redistributed away from AC1/8, whereas CaMKII continues to bind CaM_4_. This allows CaMKII to outcompete the ACs for CaM.

**Figure 7 molecules-27-05974-f007:**
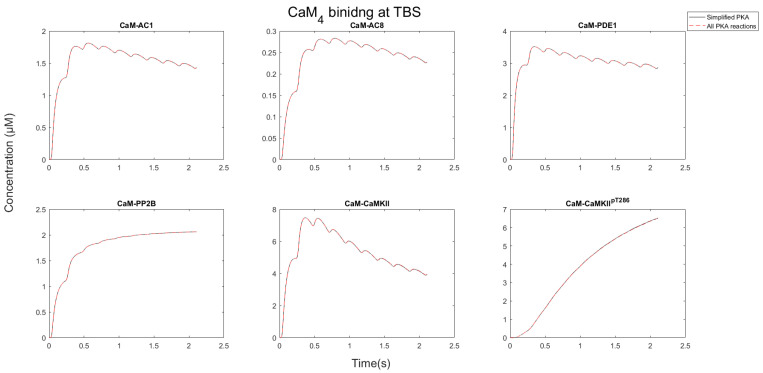
CaM_4_ binding to its partners during a TBS train. Like 100 Hz and 10 Hz, AC1, AC8, and PDE1 reduce in concentration as phosphorylated CaMKII increases. The timing of each of the 10 epochs is evident, especially for the bound portions of AC1, AC8, and PDE1 as evidenced by their peaks. Then between each peak, they dissociate from CaM_4_. The bound portions of PP2B and phosphorylated CaMKII are not subject to dissociation due to the high affinity for CaM_4_. PP2B approaches a peak concentration and then plateaus. The phosphorylated CaMKII increases linearly independent of epochs. Perhaps the function of spacing epochs is to allow for a redistribution of CaM_4_ between each epoch.

**Figure 8 molecules-27-05974-f008:**
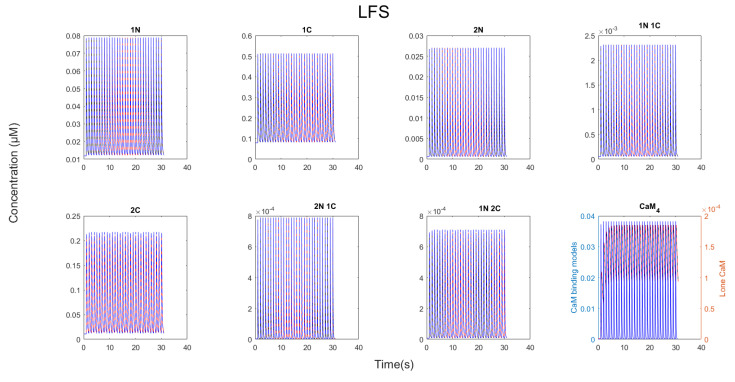
CaM states when LFS was run for 30 pulses at 1 pulse per second. There were no differences between partially bound states between the full CaM system and the non-fully bound CaM system.

**Figure 9 molecules-27-05974-f009:**
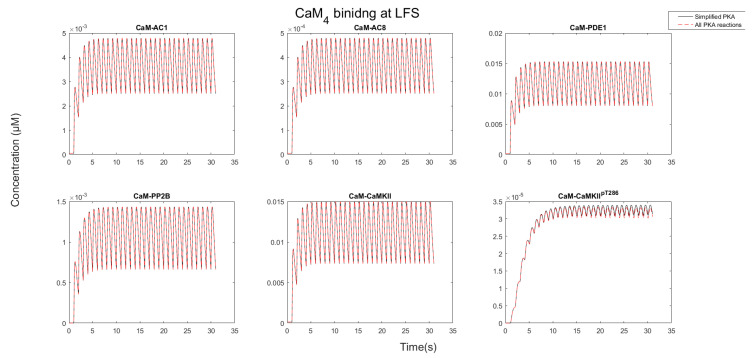
CaM_4_ bound fractions during LFS for 30 pulses. Note how PDE1 and CaMKII bound to the most but were unstable. In addition, note how CaMKII was not able to achieve stable phosphorylation. Overall, every protein is subject to much dissociation between each decaying pulse.

**Figure 10 molecules-27-05974-f010:**
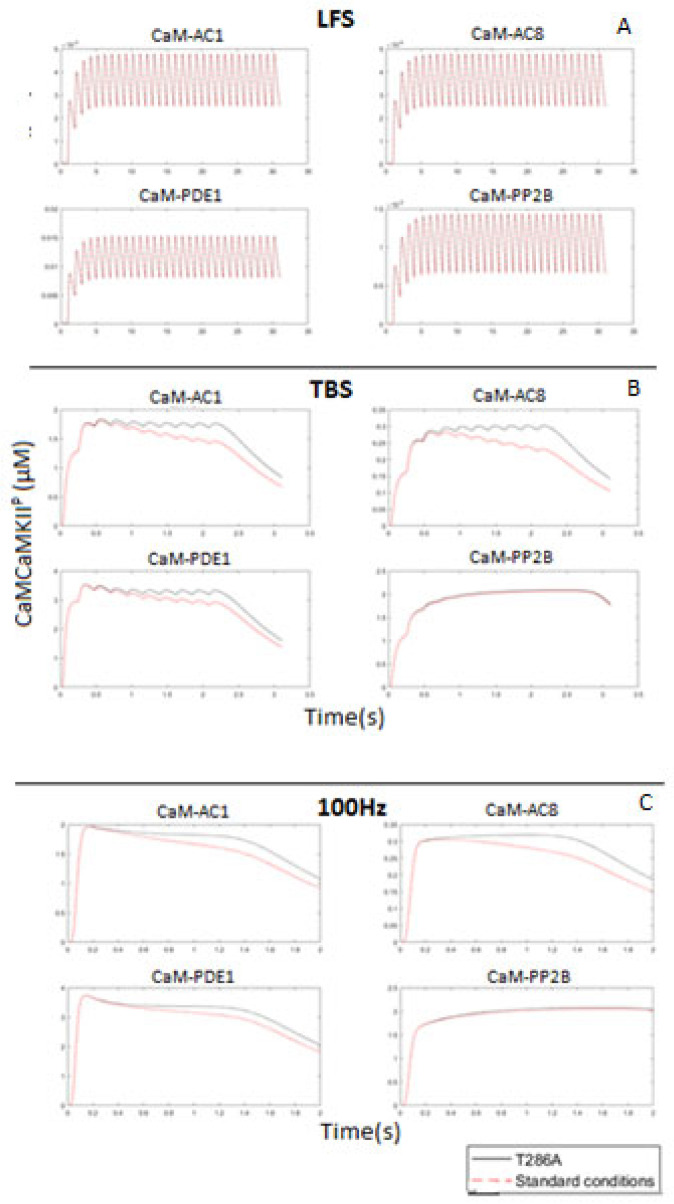
CaM bound portions of AC1, AC8, PDE1, and PP2B at LFS; 10Hz; TBS; and 100 Hz. At each stimulation, the system was run at standard model conditions or with T286 phosphorylation of CaMKII knocked out (T286A). (**A**) At LFS, there were no changes to the binding patterns. (**B**) At TBS there was an initial rise in CaM-AC1, CaM-AC8, and CaM-PDE1, followed by a decay as the stimulation progresses. (**C**) At 100Hz also there was an initial rise in CaM-AC1, CaM-AC8, and CaM-PDE1, followed by a decay as the stimulation progresses. This did not occur when T286 phosphorylation was knocked out.

**Figure 11 molecules-27-05974-f011:**
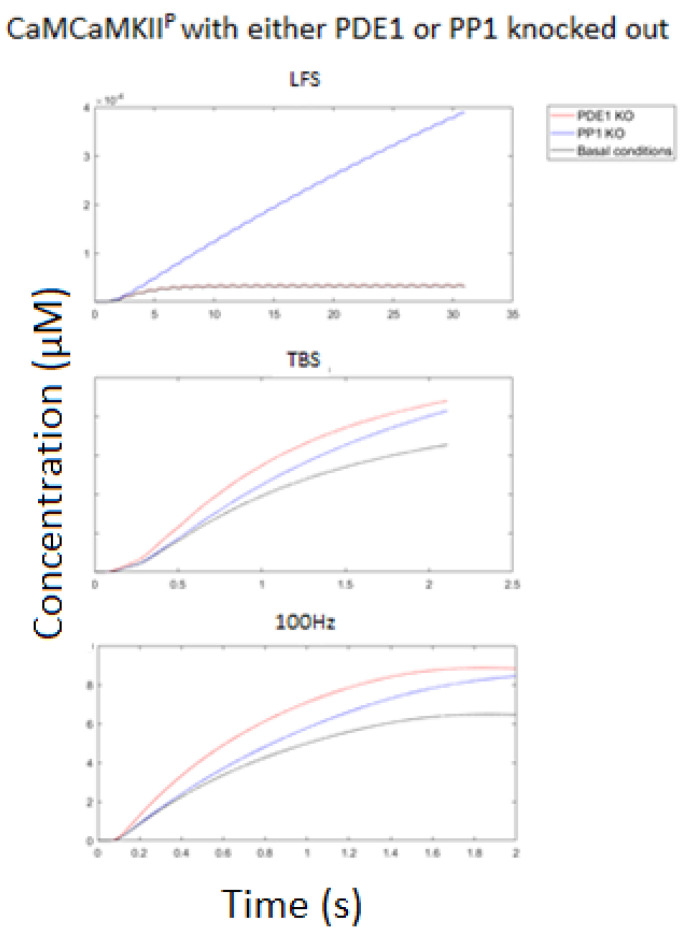
Concentration of phosphorylated CaMKII (pT286-CaMKII-CaM) at LFS, 10 Hz, TBS, and 100 Hz (from left to right) at standard conditions (black line); with no PP1 activity (PP1 KO) (blue line); PDE1 knocked out of the system (PDE1 KO) (red line). Each stimulation was run for the entirety of the stimulation then 1 s post-stimulation. Notice how pT286-CaMKII-CaM accumulated throughout the entirety of the stimulation at LFS when PP1 was knocked out, while PDE1 KO had no change versus standard conditions. At 10 Hz, knocking out PP1 and PDE1 both decreased CaMKII phosphorylation to a similar extent. At TBS and 100 Hz, knocking out PDE1 was more sensitive to CaMKII phosphorylation than PP1.

**Table 1 molecules-27-05974-t001:** CaM states in the model.

Model Variables	Variable Description
CaM_0_	calmodulin bound to no Ca^2+^ ions (apo calmodulin)
CaM_1C_	calmodulin with one Ca^2+^ ion bound to the C-lobe
CaM_1C1N_	calmodulin with one Ca^2+^ ion bound to each lobe
CaM_1N_	calmodulin with one Ca^2+^ ion bound to the N-lobe
CaM_2C_	calmodulin with two Ca^2+^ ions bound to the C-lobe
CaM_2C1N_	calmodulin with two Ca^2+^ ions bound to the C-lobe and one Ca^2+^ bound to the N-lobe
CaM_1C2N_	calmodulin with one Ca^2+^ ion bound to the C-lobe and two Ca^2+^ ions bound to the N-lobe
CaM_4_	calmodulin bound to four Ca^2+^ ions (holo-CaM)

**Table 2 molecules-27-05974-t002:** Variables in the model. The variables treated as constant and initial values are indicated. All variables are taken from Table 1 and Table 3 in [48].

Model Variable	Description	Concentration (Initial/Constant) (µM)
iAC1	inhibited AC isoform type 1	2.5 (constant)
iAC8	inhibited AC isoform type 8	0.625 (constant)
iCaMKII	inhibited CaMKII subunit	20.0 (constant)
iPDE1	inhibited PDE isoform type 1	4.0 (constant)
iPP2B	inhibited PP2B	2.1 (constant)
PP2A	protein phosphatase 2A	0.11111 (constant)
PDE4B	unphosphorylated PDE isoform 4B	1.0 (constant)
PDE4D	unphosphorylated PDE isoform 4D	1.0 (constant)
PDE4B^P^	phosphorylated PDE isoform 4B	0.0 (initial)
PDE4D^P^	phosphorylated PDE isoform 4D	0.0 (initial)
PDE4B_T_	total concentration of PDE isoform 4B	1 (constant)
PDE4D_T_	total concentration of PDE isoform 4D	1 (constant)
AC1_T_	total concentration of AC isoform type 1	2.5 (constant)
AC8_T_	total concentration of inhibited AC isoform 8	0.625 (constant)
AC*	CaM-independent AC protein(s)	2.5 (constant)
CaMKII_T_	total concentration of CaMKII	20 (constant)
PDE1_T_	total concentration of PDE isoform type 1	4 (constant)
PP2B_T_	total concentration of inhibited PP2B	2.1 (constant)
PP1_T_	total concentration of PP1	3.5 (constant)
R_2_C_2_	PKA bound to no cAMP	1.2 (constant)
R_2_C2cAMP_2_	PKA bound to 2 cAMP molecules	0.0 (initial)
R_2_C_2_cAMP_4_	PKA bound to 4 cAMP molecules	0.0 (initial)
PKAc	catalytic subunit of PKA (active)	0.0 (initial)
I1_T_	total concentration of inhibitor-1	1.5 (constant)
I1PP1	I-1-bound PP1 complex	0.0 (initial)
R_2_C_2T_	total concentration of PKA	1.2 (constant)
CaMAC1	CaM_4_ bound to AC isoform 1	0.0
CaM_T_	CaM concentration	17.7 (constant)
CaMAC8	CaM_4_ bound to AC isoform 8	0.0 (initial)
CaMCaMKII	CaM_4_ bound to CaMKII subunit	0.0 (initial)
CaMCaMKII^P^	CaMKII^P^ bound to CaM_4_	0.0 (initial)
CaMCaNA	CaNA subunit of PP2B bound to CaM_4_	0.0 (initial)
CaCaNB	Ca^2+^ bound to the small subunit of PP2B	0.0 (initial)
CaMPDE1	CaM_4_ bound to PDE isoform 1	0.0 (initial)

**Table 3 molecules-27-05974-t003:** Parameters used and values used in the full CaM model. All values used are from [48].

Parameter	Description	Value
	**Binding Rates with CaM_4_**	
kc1f	iAC1 binding CaM_4_	50 µM^−1^ s^−1^
kc1b	CaMAC1 dissociating CaM_4_	1 s^−1^
kc2f	iAC8 binding CaM_4_	20 µM^−1^s^−1^
kc2b	CaMAC8 dissociating CaM_4_	1 s^−1^
kc3f	iPDE1 binding CaM_4_	100 µM^−1^s^−1^
kc3b	CaMPDE1 dissociating CaM_4_	1 s^−1^
kc4f	CaMCaNA binding CaM_4_	46 µM^−1^s^−1^
kc4b1	CaMCaNA dissociating CaM_4_ (high Ca^2+^)	0.0012 s^−1^
K_d1_	CaMCaNA dissociation constant from CaM_4_	0.5 µM
n1	Hill constant for CaMCaNA activation	1.8
kc4b2	CaMCaNA dissociating CaM_4_ (low Ca^2+^)	2 s^−1^
K_d2_	CaMCaNA dissociation constant from CaM_4_	0.1 µM
n2	Hill constant for CaMCaNA dissociation from CaM_4_	3
kc5f	CaMKII (and CaMKII^P^) binding CaM_4_	21 µM^−1^s^−1^
kc5b1	CaMCaMKII dissociating CaM_4_	1.1 s^−1^
kc5b2	CaMCaMKII^P^ dissociating CaM_4_	0.0011 s^−1^
	**Catalytic Reactions**	
kcat1	CaMAC1 creating cAMP	2.843 s^−1^
kcat2	CaMAC8 creating cAMP	2.843 s^−1^
kcat3	AC* creating cAMP	3 s^−1^
kcat4	PKA phosphorylating PDE4B/D	18 s^−1^
Km4	Michaelis constant of PKA phosphorylating PDE4B/D	25 µM
k10	Dephosphorylation constant of PDE4B/D	0.25 s^−1^
kcat5	CaMPDE1 inhibiting cAMP	1.7 s^−1^
kcat6		3.12 s^−1^
Km5	Michaelis constant for CaMPDE1 inhibiting cAMP	10 µM
kcat7	PDE4B^P^ inhibiting cAMP	1.56 s^−1^
kcat8		3.12 s^−1^
kcat9	PDE4D^P^ inhibiting cAMP	5.4 s^−1^
kcat10		10.8 s^−1^
kcat11	Phosphorylation of I1 by PKA	1.4 s^−1^
Km11	Michaelis constant	5 µM
kcat12	Dephosphorylation of I1 by PP2A	2 s^−1^
Km12	Michaelis constant for Dephosphorylation of I1 by PP2A	16 µM
kcat13	Dephosphorylation of I1 by PP2B	2.8 s^−1^
Km13	Michaelis constant for Dephosphorylation of I1 by PP2B	3 µM
kcat14	Autophosphorylation by CaMCaMKII	1.2 s^−1^
kcat15	Dephosphorylation of (CaM)CaMKII^P^ by PP1	1.72 s^−1^
Km15	Michaelis constant for dephosphorylation of (CaM)CaMKII^P^ by PP1	11 µM
kcat16	Dephosphorylation of (CaM)CaMKII^P^ by PP2A	2 s^−1^
Km16	Michaelis constant for Dephosphorylation of (CaM)CaMKII^P^ by PP2A	16 µM
	**cAMP Binding**	
k17f	2 cAMPs binding R_2_C_2_ (PKA)	8 µM^−2^ s^−1^
k17b	2 cAMPs dissociate from R_2_C_2_ (PKA)	0.02 s^−1^
k18f	2 cAMPs binding R_2_C_2_ cAMP_2_ (PKA)	0.7 µM^−2^ s^−1^
k18b	2 cAMPs dissociate from R_2_C_2_ cAMP_2_ (PKA)	0.2 s^−1^
k19f	2 catalytic domains of PKA (PKAc) become active	0.25 µM^−2^ s^−1^
k19b	PKAc rebinds regulatory domain	0.0016 s^−1^

**Table 4 molecules-27-05974-t004:** GSA of CaM4. Each row represents a model parameter and its PRCC at each of the three stimulations. Empty cells represent no significant PRCC value (*p* < 0.0001). The color indicates the significance of PRCC values: Blue is low significance; Green is moderate significance; Yellow is high significance; and Red means very strong significance.

	LFS	TBS	100 Hz
kon_NT_	0.309447	0.520403	0.42571
kon_NR_	0.338254	0.531797	0.485552
koff_NT_	0.287282	0.491028	0.390258
koff_NR_	0.350848	0.531098	0.498465
kon_CT_	0.316173	0.320826	
kon_CR_	0.333078	0.392427	0.194096
koff_CT_	0.333058	0.283302	
koff_CR_	0.234501	0.269792	0.127361
K1	0.184933		
kcat3	0.206065		
kcat5	0.095151		
kcat11	0.205424		
kcat14	0.174413	0.236121	0.142064
kcat15	0.265419	0.141936	
kcat16	0.059605		
Km5	0.07769		
Km9	0.087172		
Km11	0.121534		
Km15	0.271254	0.093537	
Km16		0.067706	
kc1b		0.066082	0.100113
kc3f		0.061195	
kc3b		0.065239	
kc5f	0.314694	0.459728	0.457296
kc5b1	0.294508	0.378356	0.371833
PKAmax	0.211577		
AC1T		0.116044	0.202595
CaMKIIT	0.124258	0.746962	0.826951
PDE4DT	0.210772		
“PDE1T”	0.412729	0.212812	0.288863
PP2BT	0.181503	0.186437	0.203467
PP2AT	0.213967		
PP1T	0.191032	0.142581	
I1T	0.303947		
Total Number	29	22	14

**Table 5 molecules-27-05974-t005:** GSA of cAMP. Each row represents a model parameter and its PRCC at each of the three stimulations. Empty cells represent no significant PRCC value (*p* < 0.0001). The color indicates the significance of PRCC values: Blue is low significance; Green is moderate significance; Yellow is high significance; and Red means very strong significance.

	LFS	TBS	100 Hz
kon_NT_	0.089041		
kon_NR_	0.094016	0.10346	
koff_NT_	0.145789	0.0772	
koff_NR_	0.117678		
kon_CT_	0.105949		
kon_CR_	0.14284		
koff_CT_	0.096766		
koff_CR_	0.099329		
K1	0.486762	0.134202	
Kcat1	0.115366	0.676005	0.672556
Kcat2			0.188206
Kcat3	0.555548		
Kcat5	0.151363		0.120325
Kcat6		0.099481	0.221786
Kcat7	0.083178		
Kcat9	0.390133		
Kcat14		0.104717	0.077823
Kcat15			
Km5	0.179027	0.081035	
Km7	0.094523		
Km9	0.440493		
kc1f	0.098626	0.338759	0.56966
kc1b	0.096008	0.323026	0.247466
kc2f		0.117612	0.128951
kc2b		0.092723	0.075022
kc3f			0.116261
kc3b			
kc5f		0.28144	0.501796
kc5b1		0.203474	0.395388
k17f	0.404247	0.823731	0.572203
k17b	0.226583	0.178773	
R_2_C_2_T	0.084222	0.129164	0.426912
AC1T	0.106218	0.661426	0.673051
AC8T	0.005379	0.181439	0.176803
CaMKIIT		0.469565	0.49394
PDE4BT			0.079848
PDE4DT	0.399932	0.084721	0.1995
PDE1T	0.112717	0.236887	0.510997
PP2BT		0.083351	0.102463
AC*T	0.559471	0.153316	
Total Number	27	23	21

**Table 6 molecules-27-05974-t006:** GSA for I1PP formation. Each row represents a model parameter and its PRCC at each of the three stimulations. Empty cells represent no significant PRCC value (*p* < 0.0001). The color indicates the significance of PRCC values: Blue is low significance; Green is moderate significance; Yellow is high significance; and Red means very strong significance.

	LFS	TBS	100 Hz
kon_NT_	0.087518	0.092052	
kon_NR_	0.089937	0.096011	
koff_NT_	0.127441	0.094289	
koff_NR_	0.107561		
kon_CT_	0.089898	0.114253	
kon_CR_	0.147144	0.134583	
koff_CT_	0.092159	0.125213	
koff_CR_	0.10766		
K1	0.611403		
Kd1	0.122036		
Kcat1	0.110938		
Kcat3	0.641382		
Kcat5	0.161092		
Kcat7	0.104808		
Kcat9	0.469457		
Kcat11	0.324548	0.174653	0.091772
Kcat12	0.081627		
Kcat13	0.104938	0.788718	0.82291
Km5	0.190914		
Km7	0.090352		
Km9	0.505001		
Km11	0.23138	0.163702	
Km13	0.091816	0.72597	0.749709
kc1f	0.10949		
kc1b	0.115088		
kc4f		0.365372	0.377969
kc5f		0.152477	0.162573
k17f	0.094235		
k18f	0.277112		
k18b	0.235321		
k19b	0.288588		
R_2_C_2_T	0.13655		
AC1T	0.118128		
CaMKIIT		0.177128	0.314058
PDE4DT	0.463679		
PDE1T	0.130136		
PP2BT		0.776154	0.80817
I1T	0.236879	0.163962	
AC*T	0.639797		
Total Number	35	15	7

## Data Availability

Not applicable.

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
