# Peer review of "A Modeling and Analysis Study Reveals That CaMKII in Synaptic Plasticity Is a Dominant Affecter in CaM Systems in a T286 Phosphorylation-Dependent Manner"

_molecules, 2022, doi:10.3390/molecules27185974_

Round 1

Reviewer 1 Report

This is a potentially interesting study that models the dependence of the synaptic potentiation / depression (LTP/LTD) balance in dependence of calmodulin activation by calcium and the calmodulin concentration in neurons. Calmodulin-activated enzymes, in particular CaMKII, are major calcium sensing proteins in neurons, that link neuronal activity levels (via depolarization and calcium influx) to biochemical signaling pathways. Therefore, this work is of interest.

The manuscript can be improved when achieving a somewhat elevated level of accuracy and precision. This folds for both, the presentation of the scientific premise as well as the model itself. With regard to the introduction and background information, it is a commonly held notion that generally high calcium signals cause LTP via activation of CaMKII, whereas low (sometimes prolonged) calcium signals cause LTD via activation of phosphatases (the only directly calcium-calmodulin activated phosphatase in neurons is PP2B). Nevertheless, this notion is not accurate as it cannot be generalized. Work on cerebellar plasticity has shown that at cerebellar parallel fiber to Purkinje cell synapses high calcium (still via CaMKII) induces LTD, while low calcium (via PP2B) induces LTP. This should be noted for accuracy. A good reference is Jorntell and Hansel, Neuron 52, 2006.

As for the model, it seems that omission of factors influencing CaMKII activity could lead to very different outcomes. Two factors should be added to the model, or at least the omission should be discussed as a caveat:

1)    The authors focus on the interaction of CaMKII with PP1. In terms of competition for calcium-activated calmodulin, PP2B is more relevant (see Lisman’s work). This should be highlighted more strongly.

2)    Inhibitory autophosphorylation at T305/306 does not seem to be considered. This is a concern as not sufficiently long/strong CaM binding will reduce CaMKII activity (directly influencing the outcome of the modeling presented here) and has been shown to affect hippocampal plasticity (Elgersma et al., Neuron 36, 2002) as well as the shifting of LTP/LTD induction thresholds (Piochon et al., PNAS 113, 2016). This is an aspect that should at least be discussed in a separate paragraph.

Reviewer 2 Report

Why the results of the study were mentioned in the introduction section?

2. Table 2 was not mentioned in the text, and further explanation regarding the

concentrations needs to be added.

3. What is the explanation of this note? “Notice that each partially bound states

peak and rapidly decays in the full CaM binding system, unlike in the lone

CaM system.” in Figure 5. Discuss it, and why?.

4. The fourth point in the results should be discussed previously in the

methods.

5. You need to mention a statistical analysis section in the methods as you

mentioned the p values in your tables.

6. Minor: Edits

In line 152 phenomologically

In line 164 indicates  indicate

In line 189 there is a space before These.

In line 225 in the table, there is extra space before Dephosphorylation

In line 505 (has little influence) is repeated

Round 2

Reviewer 1 Report

In their response, the authors state that the suggested references were cited. That is not correct for Piochon et al, PNAS 113 (2016). The authors should discuss this paper in the new paragraph on inhibitory autophosphorylation and sliding thresholds.

Author Response

Thanks for the comments. A new paragraph very briefly discussing the paper, Piochen et el. (2016), is added as suggested. See light blue highlighted lines from 121-129.

The manuscript was checked spelling and grammar using Word speller and grammar check.

Reviewer 2 Report

The authors made all the required modifications. 

Author Response

No further comments to reply.